# OpenReview forum: "SPRING: Studying Papers and Reasoning to play Games"
_NeurIPS.cc/2023/Conference — NeurIPS 2023 poster_

### Official Review · Reviewer_FDDh · 2023-06-29

**Soundness:** 4 excellent
**Presentation:** 4 excellent
**Contribution:** 3 good
**Rating:** 7
**Confidence:** 3

**Summary:**

Open-world survival games such as Crafter pose a significant challenge for RL agents. In this paper, the authors propose a novel framework called SPRING that studies Crafter's latex paper source code and uses the knowledge to reason and play the game through large language models (LLMs) such as GPT4. SPRING has two phases:

1. Phase 1 is studying the paper. SPRING extracts key information relevant to the game from the LaTeX source code, and summarized them in bullet list forms.
2. Phase 2 is reasoning and playing the game. SPRING uses the summarized information in Phase 1 as the context of the LLM, and formulates gameplay-relevant questions as a directed acyclic graph (DAG). When traversing through the DAG, the agent only looks at the questions and answers in the parent nodes to reduce context length which helps the agent focus better. After the DAG of questions the agent finally outputs an action to be executed in the game.


The authors then did a comprehensive ablation study showing that 1) using the DAG of questions is important, 2) using extracted / cleaned information from the paper is important, and 3) using GPT-4 is much better than using GPT-3.5

**Strengths:**

The paper is well-written and presents a carefully designed ablation study. The evaluation solidly demonstrated that the two phases of SPRING contributed to a higher reward. The approach to use LM is also highly novel in open-world survival games.

**Weaknesses:**

The paper is quite well-done, but maybe one weakness is its reliance on proprietary models such as GPT4. In 5 years or so, it is unclear whether OpenAI will still make the (then probably deprecated) GPT4 available, so this work could become more difficult to reproduce.

**Questions:**

Could you include some gameplay and output examples in the appendix? They could help the readers better understand the flow of the system. Could you also disclose the computational cost associated with GPT-4 API calling used in the paper?

**Limitations:**

Yes.

---

> ### Author Rebuttal · Authors · 2023-08-09
>
> Thank you for recognizing the solidness of our work. Here are our responses to the questions and concerns:
>
>
> W1 proprietary models:
>
> Thank you for pointing out this important concern. We do notice that SPRING gets better after switching from GPT-4-0314 to GPT-4-0613, and we are hopeful that we get better performance with every new version. However, we are also actively experimenting with open-source LLMs like LLAMA-2 and we are hopeful to make a switch to an open-source LM in the future.
>
>
> Q1 Examples and Gameplay:
>
> We will include some rollout examples of SPRING in the final paper.
>
> The number of queries per step is 9 (same as the number of questions). Each game can take around 300 steps, but can go up to 500 steps in the worst case. Therefore, the maximum number of queries per game can go up to 4500. According to the public price of GPT-4 API, each query costs around 0.06. The total cost should be less than 270 (USD) with GPT-4. Note that GPT-4 cost will be cheaper with academic discounts and we expect the cost to decrease as the community makes progress on LLMs.

---

### Official Review · Reviewer_AJEx · 2023-07-05

**Soundness:** 2 fair
**Presentation:** 2 fair
**Contribution:** 2 fair
**Rating:** 5
**Confidence:** 4

**Summary:**

This paper proposes SPRING to read the game’s original academic paper and use the knowledge learned to reason and play the game throught a large language model (LLM).

**Strengths:**

1. This paper is well-writen and easy to follow.
2. The idea of reading paper with LLM to interact with environment directly is interesting.

**Weaknesses:**

1. Minecraft's open-world nature, extensive online community discussions (e.g., Reddit) and comprehensive wikis raise questions about why SPRING chose to use academic paper latex code for prompts instead of utilizing more abundant online corpora. It would be necessary to conduct a suitable ablation experiment.
2. The experiments conducted in the paper are insufficient. Currently, the paper only tests SPRING in the simplified 2D world of "crafter" with a modified action space. It is essential to investigate SPRING's performance in a real open-world environment, as the paper frequently references Minecraft. Experiments should be conducted in environments like Minedojo or Minerl.
3. Directly using LLM for action output may raise concerns about efficiency. Given the current parameter scale of language models and the multi-stage question-answering (QA) used by SPRING, can SPRING achieve real-time action output at 20fps in the crafter environment? Merely listing training steps is insufficient; please provide the average inference time consumption for all methods in Table 2.
4. Although the crafter environment itself is not part of GPT-4's training data, the rules used in crafter are fundamentally similar to Minecraft (e.g., using a pickaxe to mine stone), making it challenging to claim that the SPRING method is genuinely out-of-distribution (OOD).
5. In the QA-DAG section, the paper employs multiple manually designed questions as nodes, some of which may leak game rules, such as q1, q3.

**Questions:**

Please address the questions raised in the weaknesses section.

---

> ### Author Rebuttal · Authors · 2023-08-09
>
> Thank you for recognizing the novelty of our work. Here are our responses to the questions and concerns:
>
>
> W1 Online Copra:
>
> No other online corpora exist for crafter since it is an academic benchmark environment. This benchmark also provides us the chance to test our framework on unseen data. Data on Reddit and online corpora are most likely already seen by GPT-4 during its training process.
>
>
> W2 Insufficient Evaluation:
>
> Thank you for this insight and we plan to add Minedojo as a part of future work. However, given the fact that it is significantly harder to 'describe' Minecraft environments to LLMs, we make the decision to start with a simplified environment which abstracts out some complexity on the vision side.
>
> Although our ultimate goal is to solve Minecraft, we are excited to share our progress with the community. We would like to re-iterate that Crafter greatly resembles Minecraft in the tech-tree aspect and our work shows that LLMs can understand and excel in games with complex tech-trees, which is a first step towards solving Minecraft.
>
>
> W3 Inference Speed:
>
> Current inference time depends on the speed of the OpenAI API, which involves HTML and therefore has slower inference time than some locally trained RL algorithms depending on the local machine capability. However, this also suggests that our inference can be done on any device (even raspberry pi) with an internet connection. Therefore, we don't believe a comparison would be helpful for understanding the different algorithms.
>
> However, we do recognize the necessity to reduce API calls for LLMs, and we are actively working on a more API-efficient solution as our future work.
>
>
> W4 OOD:
>
> Thank you for raising the concern. We will change the wording from OOD to 'unseen'.
>
> In addition, please note that GPT-4 encounters difficulties playing the game of Crafter without reading the paper (Table 3 row 4).
> GPT-4 encounters more 'hallucinations' without the context due to the fact that rules are fundamentally similar to Minecraft.
>
>
> W5 prompts:
>
> Thanks for raising the concerns. We will have GPT-4 write the questions for our future works.

---

> > ### Author Response · Authors · 2023-08-17
> >
> > Dear reviewer,
> >
> > Thank you again for helping us improve our paper. We hope that our response has addressed your concerns.
> >
> > Please let us know if there are any additional questions or concerns.

---

> > > ### Comment · Reviewer_AJEx · 2023-08-18
> > >
> > > Thank you for the author's reply. I will still maintain my current score.

---

### Official Review · Reviewer_XAmG · 2023-07-06

**Soundness:** 4 excellent
**Presentation:** 3 good
**Contribution:** 4 excellent
**Rating:** 8
**Confidence:** 4

**Summary:**

The authors propose a method, SPRING, to design an agent based on LLM question answering for the Crafter environment. Their method uses the Hafner 2021 paper describing the Crafter environment and its objectives as context to prompt the LLM about the next best steps in a chain-of-thought fashion using a DAG to organise the sequence of prompts, based on text descriptions of the environment provided by an image to text visual descriptor. SPRING outperforms other approaches, including some using LLMs as components.

**Strengths:**

1. A logical and interesting approach based on having Hafner 2021 as a good “game guide” for the player. Agent modelling with LLM has been a promising area of research, so I expect this paper to be of interest to a large audience. Thanks to the question-answering and chain-of-thought approach, this approach provides some explainability. What’s more, the authors show that their chain-of-thought approach with the DAG improves on other forms of chain-of-thought, highlighting the importance/impact of guiding the chain-of-thought process with their methodology.
2. The writing is good and clear.
3. I appreciate the authors keeping perspective by using quotation marks for reasoning.


**Weaknesses:**

1. The comparison to RL algorithms had to be done. However, it is important to keep in mind that SPRING is operating with explicit knowledge of “solutions”, or at least a guide, which is further assisted by the chain-of-thought prompting with the DAG. In a game where the difficulty is not mechanical, and which is about finding the right steps, at the risk of anthropomorphising the algorithm, it sounds clear to me as a human that having the details from Hafner 2021 “cheeses” the game. Pure RL agents in comparison have much more to learn by themselves. A corollary is that I might trust RL agents more for generalisation, due to exploration (the total failure of “place stone” is a good indication that the model is just following Hafner 2021 as a guide). This operates at two levels: your method has access to explicit formulations of the objectives and rewards that your benchmark is based on, and it has access to a guide on how to accomplish said objectives.
2. I would replace the claim L47 that the environment is OoD. Being unseen doesn't mean it's out of distribution, I would correct it to "unseen". For example a model trained on StarCraft 1 would probably be decent at StarCraft 2 because StarCraft 2 corresponds to a distribution that's shifted but probably overlaps to a large extent with that of StarCraft 1. Same for a model trained on shooter games. It may very well be the case here that a lot of knowledge from the LLM pretraining transfers well (again at the risk of anthropomorphising the model, but think about a player used to similar video games as crafter vs someone who never played a video game. Wouldn’t the video-game aficionado progress much faster ? That would be mostly due to the fact that past experience transfers in spite of the environment being unseen).
3. Table 3: if possible, please indicate scores of each modality too.
4. Abstract: in summarising your results, please specify you’re evaluating on Crafter (I assume you intended to specify it originally, as otherwise the sentence “We propose a novel approach, SPRING, to read the game’s original academic paper” lacks meaning). This is important since your methodology is tailored to Crafter, using Hafner 2021 with all benchmarks being on Crafter.

I believe adding a discussion of 1. for SPRING vs other baselines would be very valuable to readers.

**Typos and suggestions:**
1. Missing whitespace in abstract between node and directly
2. L64 "independently in."
3. L69 "performance performance"
4. Figure 3: Grass 1 step no s
5. L136 simpler instead of more simple
6. L196 “achives”
7. L306: “reliance of” -> on
8. Table 2: Just a remark, not affecting my judgement: it’s important to keep in mind that the improvement in reward over DreamerV3 is not statistically significant.
9. Suggest using citep for references that aren't grammatical parts of sentences, will improve legibility (e.g. “It has been done previously [Whatever et al.]” instead of “It has been done previously Whatever et al”).


**Questions:**

1. Will you be providing code for reproducibility ?
2. What is the success rate computed from in 3.2? Multiple runs?
3. It is not clear whether Figure 4 of Hafner 2021 is read by the LLM somehow (specifically the arrows indicating achievement orders). Can you clarify how explicitly the LLM is provided with the tech tree dependencies please ? Is there another source for that knowledge than Figure 4 of Hafner 2021 (because surely the tokeniser can’t process the arrows) ?


**Limitations:**

Yes.

---

> ### Author Rebuttal · Authors · 2023-08-09
>
> Thank you for appreciating our chain-of-thought approach and recognizing the potential of our work. Here are our responses to the questions and concerns:
>
> W1 Game complexity:
>
> Thank you for pointing out an important aspect that the Crafter lacks mechanical difficulty, which is indeed the intended goal of our paper. However, this does not mean that the game would be obvious for human players who read the paper carefully. The instructions (Hafner et al.) only cover the high-level game mechanics, i.e., the tech-tree and the enemies. It is unclear where to find the resources and when you would encounter a zombie/skeleton.  The exact speed and movements of important objects: cow, zombie, skeleton, arrows (shoot by skeleton) are not documented in the paper. In fact, the paper did not even mention that 2 woods (as opposed to 1) are required to build a crafting table. All of the above missing low-level details still pose challenges to LLMs and novice human players.
>
> In addition, Hafner et al. only demonstrated the existence of many conflicting objectives like: "Do I explore or build shelter.", "Do I focus on survival stats or do mining?", but never provided a solution. Our experience is that it may take a human player several trials to reach a score higher than our LLM due to some of the difficulties mentioned above.
>
> In regards to SPRING vs. RL in the generalization setting. We think that it is hard to say if one is better than the other.
> On one hand, RL algorithms are trained with reward functions deliberately engineered to cover all in-game achievements. Engineered reward functions often require a lot of expert knowledge and careful formulation.
> On the other hand, SPRING does not need the reward (we report reward for comparison purpose, SPRING does not use the reward during inference), but instead uses external knowledge (Hafner et al.).
>
> We will include a discussion of SPRING vs. RL in the final paper. However, we will not be able to include more insights (than Table 3, row 2) on how LLM adapts to some of the missing low-level inaccuracies since we are still actively working in this direction.
>
>
> W2 OOD:
>
> Thank you for pointing out this inaccuracy. We will change our phrasing of OOD to unseen in the final version
>
>
> W3 Multi-modality:
>
> Apologies for any potential misunderstanding. Do you mean removing the visual modality? The visual modality (visual descriptor) is necessary for the agent. Without the visual modality, it will not know where the resources are and if it is facing any danger, which are fundamental to success in the game.
>
>
> W4 Abstract:
>
> Thank you for pointing out this inaccuracy. We will update the abstract to explicitly mention that we only benchmark on crafter in the final version.
>
>
> Misc:
>
> Thank you for pointing out minor improvements in our paper. We will correct them in the final version.
>
>
> Q1 Code:
>
> Yes, we will release the code upon publication.
> In addition, we have already attached ALL prompts used in the main paper.
>
>
> Q2 Success Rate:
>
> Success rate is calculated as (number of times an achievement is achieved)/(total number of experiments). In our case the total number of experiments is 5.
>
>
> Q3 Figure 4 from (Hafner et al.):
>
> Figure 4 is not read since the format is not in LaTeX.
>
> However, the same information is also in Appendix Table F.1 of the paper which is in a fully readable LaTeX table format.
>
> We will add clarification to this confusion in the final version of our paper.

---

> > ### Comment · Reviewer_XAmG · 2023-08-14
> >
> > I would like to thank the authors for their detailed reply.
> >
> > > All of the above missing low-level details still pose challenges to LLMs and novice human players.
> > In addition, Hafner et al. only demonstrated the existence of many conflicting objectives like: "Do I explore or build shelter.", "Do I focus on survival stats or do mining?", but never provided a solution. Our experience is that it may take a human player several trials to reach a score higher than our LLM due to some of the difficulties mentioned above.
> >
> > Reviewing aside, I would argue that this is precisely because a human would try exploring "policies", which will only pay off in the longer run. I agree that reward modelling is where a lot of the complexity of RL gets pushed.
> >
> > > Apologies for any potential misunderstanding. Do you mean removing the visual modality? The visual modality (visual descriptor) is necessary for the agent. Without the visual modality, it will not know where the resources are and if it is facing any danger, which are fundamental to success in the game.
> >
> > Sorry, I meant for each method.
> >
> > I have increased my score (7 -> 8).

---

> > > ### Author Response · Authors · 2023-08-16
> > >
> > > Thank you for the recognition of our work.
> > >
> > > - Exploration.
> > >
> > > Thank you for identifying this important direction. As a future direction, we have been studying "exploration" with LLMs similar to how a human player would explore and learn.
> > >
> > >
> > > - Score for Table 3
> > >
> > > For Table 3, we made the decision to not calculate score since the scores were designed for benchmarking the exploration (how many achievements an agent unlocks) and training speed (when does an agent unlock achievements) of RL policies.
> > > None of which are relevant for LLM-based policies which were not trained on Crafter. Therefore, we instead focus on metrics which demonstrate achievement sophistication (achievement depth), and achievement amount (reward).
> > >
> > > Thank you for pointing out this omission, we will add clarification for our choice of metrics in the final version of the paper.

---

### Official Review · Reviewer_QUqN · 2023-07-07

**Soundness:** 3 good
**Presentation:** 3 good
**Contribution:** 3 good
**Rating:** 6
**Confidence:** 3

**Summary:**

SPRING is a method that reads a game (Crafter)'s original academic paper and reasons to play the game with a large language model (LLM).
1. study the paper: a fixed set of questions are used to summarize gameplay and action space information, with parapraph-level decomposition and then question-wise aggregation.
2. reasoning: a fixed list of 9 questions in a DAQ are answered by LLM using game information (described in text) and answers from parent questions.

Results are impressive --- such a zero-shot model beats RL baselines with 1M training steps, also with a smaller variance.

**Strengths:**

- It is an important and popular direction now to study how to implement RL agents using LLM, so as to avoid typically millions of training steps and problems like generalization and robustness. The paper makes great progress in this direction, significantly beating RL baselines in the game of Crafter.

- Components are reasonable and fairly novel. Can imagine these techniques be useful for other games as well.

- Good component and ablation studies, and good presentation.

**Weaknesses:**

- Since only one game is studied, it is harder to understand how much of performance comes from many human inductive bias in designing the pipeline (e.g. questions chosen, descriptor ad-hoc rules, etc.)  --- these things might be strong enough priors that LLM just serves as a symbolic processor, and might even be replaceable with some simple rules. It would be cool (but hard) to see to what extent LLM itself can just be also replaced by some heuristic rules to solve the game --- and if it performs poorly, the use of LLM (and hardness of game) is more justified.

- I'm a bit surprised (maybe not that surprised) "Learning to Win by Reading Manuals in a Monte-Carlo Framework" is not cited, which is probably the groundbreaking work in this direction from 12 years ago.

**Questions:**

- what's the cost/speed for GPT-4 experiments?

- is there any statistics for RL performances beyond 1M steps, which might not be trained enough?

- typo: line 81 "key aspects the visual observation"; line 64 "independently in"

- use \citep

**Limitations:**

limitation about visual descriptor is mentioned.

---

> ### Author Rebuttal · Authors · 2023-08-09
>
> Thank you for recognizing the novelty and importance of the direction of our work. Here are our responses to the questions and concerns:
>
>
> W1 Rule-based agents:
>
> Please note that the complexity of the game often goes beyond human inductive bias. The game involves a tech-tree that resembles Minecraft. It is unclear if a rule-based agent exists as there are often conflicting objectives like: "Do I explore or build shelter.", "Do I focus on survival stats or do mining?"
>
> If huge effort can be put into writing a rule-based agent, then similar efforts might be used to improve the context prompt to the LLM (which is currently automatically extracted straight from the paper)
>
> We will provide more example trajectories of the game in the appendix to demonstrate the complexity for the final version.
>
>
> W2 Missing citations:
>
> Thank you for pointing out the missing citation. We will add this to our final version.
>
>
> Q1 Cost/Speed of LLM:
>
> The number of queries per step is 9 (same as the number of questions). Each game can take around 300 steps, but can go up to 500 steps in the worst case. Therefore, the maximum number of queries per game can go up to 4500. According to the public price of GPT-4 API, each query costs around 0.06. The total cost should be less than 270 (USD) per game with GPT-4. We would also like to point out that as this area of research is becoming increasingly popular, the cost per query is becoming cheaper and the inference is becoming faster.
>
> The actual speed of the agent largely depends on the speed of the LLM API.
>
>
> Q2 1M training steps:
>
> The 1M steps is a rule set by the original game developers (Hafner et al.) https://arxiv.org/pdf/2109.06780.pdf. Therefore, no scores exist for beyond 1M steps.
>
>
> Misc:
>
> Thank you for pointing out minor improvements in our paper. We will correct them in the final version.

---

> > ### Comment · Reviewer_QUqN · 2023-08-14
> > **Thanks**
> >
> > Thanks. I want to keep the score as is for now.

---

### Decision · Program_Chairs · 2023-09-21

**Decision:**

Accept (poster)

**Comment:**

This paper proposes a novel approach called SPRING to use LLMs for playing Crafter. The approach involves adding the Crafter paper to the context, asking targeted questions from LLMs about it and glueing the answers in a specific way. One of the main interesting points from the paper is to demonstrate that how powerful LLMs are when prompted the right way. On the other hand, the questions, reasoning module and other choices are very specific to this game which makes it difficult to use for other environments. That being said, this is a great first step towards general solutions that employ LLMs effectively for this purpose. All reviewers voted for accepting this paper. Therefore, I recommend acceptance.